# Rosmarinic Acid-Grafted Dextran/Gelatin Hydrogel as a Wound Dressing with Improved Properties: Strong Tissue Adhesion, Antibacterial, Antioxidant and Anti-Inflammatory

**DOI:** 10.3390/molecules28104034

**Published:** 2023-05-11

**Authors:** Yi Yin, Qianqian Xu, Xin Wei, Qianyun Ma, Dongsheng Li, Juanjuan Zhao

**Affiliations:** 1School of Pharmacy, Binzhou Medical University, Yantai 264003, China; 2Tianjin Key Laboratory of Innovative Ophthalmic Optics Technology, Tianjin Shiji Kangtai Biomedical Engineering Co., Ltd., Tianjin 300462, China

**Keywords:** hydrogel for wound dressing, rosmarinic acid, strong tissue adhesion, antibacterial, antioxidant, anti-inflammatory

## Abstract

Designing a strong tissue adhesive and multifunctional hydrogel dressing for various skin injuries is still a significant challenge. Based on the bioactive activities of rosmarinic acid (RA) and its catechol structure being similar to dopamine, RA-grafted dextran/gelatin hydrogel (ODex−AG−RA) was designed and systemically characterized in this study. The ODex−AG−RA hydrogel exhibited excellent physicochemical properties, including fast gelation time (61.6 ± 2.8 s), strong adhesive strength (27.30 ± 2.02 kPa) and enhanced mechanical properties (1.31 × 10^4^ Pa of G′). The examination of hemolysis and co-culturing with L929 cells showed the strong in vitro biocompatibility of ODex−AG−RA hydrogels. The ODex−AG−RA hydrogels exhibited a 100% mortality rate against *S. aureus* and at least 89.7% against *E. coli* in vitro. In vivo evaluation for efficacy in skin wound healing was carried out in a rat model of full-thickness skindefect. The amount of collagen deposition and CD31 on wounds in the two ODex−AG−RA−1 groups on day 14 was 4.3 times and 2.3 times of that in the control group, respectively. Furthermore, the mechanism of ODex−AG−RA−1 for promoting wound healing was proved to be related to its anti-inflammatory properties by adjusting the expression of inflammatory cytokines (TNF-α and CD163) and reducing the level of oxidative stress (MDA and H_2_O_2_). Overall, this study demonstrated the wound-healing efficacy of RA-grafted hydrogels for the first time. ODex−AG−RA−1 hydrogel, due to its adhesive, anti-inflammatory, antibacterial and antioxidative activities, was a promising candidate as a wound dressing.

## 1. Introduction

Skin, the largest organ of the human body, plays an important role as a barrier [1,2]. However, skin injuries such as burns, cuts, chemical injury or other damages are a common problem in daily life and can lead to infection [3,4,5]. Skin injuries damage the integrity of the barrier and expose tissue to the external environment. Thus, the possibility of microbial infection increases and wound healing slows [6,7]. Therefore, how to promote fast and effective wound healing is still a research focus in clinical practice.

Wound dressing is considered to be an effective method of promoting healing in clinical practice [8]. Hydrogels, which are composed of hydrophilic three-dimensional networks similar to the natural extracellular matrix, have been regarded as the most promising modern dressing [9,10]. Due to their hydrophilic nature, hydrogels can not only absorb wound exudate and maintain a moist environment for wounds, but also reduce the risk of wound infection and accelerate wound healing. Various hydrogels based on natural polymers such as dextran, chitosan, polysaccharide, collagen and gelatin have been developed due to their non-immunogenicity and biocompatibility [11,12,13]. Dextran is a naturally colloidal, hydrophilic, biocompatible and nontoxic polysaccharide and has been used in biomedical fields for years [14]. Gelatin, a collagen-hydrolyzed protein, has many characteristics including non-immunogenicity, cell adhesion behavior and blood coagulation. Thus, gelatin is believed to promote wound healing [15]. Therefore, dextran or gelatin has been used in combination with other synthetic or natural macromolecules to fabricate wound dressings. However, hydrogel dressings based on common dextran or gelatin suffer from some shortcomings, such as weak tissue adhesion, low mechanical strength, undesirable degradation rate and lack of functionality. Herein, multiple chemical or physical methods to modify dextran or gelatin dressings have been studied to develop appropriate hydrogel dressings. Catechol groups, which were inspired by mussels, have attracted much attention recently and been grafted to hydrogels to enhance tissue adhesiveness [16,17]. Usually, the catechol groups were provided by dopamine. Although hydrogels grafted to dopamine had better tissue adhesion, their medical functions, such as anti-inflammatory, antioxidant and antibacterial activities, have not been much improved. Therefore, designing a multifunctional hydrogel dressing is still significant for various skin injuries.

The inflammatory reaction is the first stage of wound healing, and moderate inflammation could resist infection and promote healing [18,19]. However, the prolongation of the inflammatory reaction aggravates tissue damage and ultimately affects wound healing [20]. Additionally, the increased reactive oxygen species (ROS) produced in wound healing could delay the healing process by changing or degrading extracellular matrix (ECM) proteins, damaging dermal fibroblasts and reducing the function of keratinocytes [21,22]. Moreover, it is well known that bacterial infection is very harmful to wound healing. Thus, hydrogels with the ability to regulate the inflammatory reaction and antioxidant and antibacterial activities could effectively promote wound healing.

Bioactive compounds derived from herbal plants have been used in the pharmaceutical industry, tissue engineering, meal additives and other biomaterials [23]. Among such natural chemical compounds, phenolic components have attracted great attention and proven beneficial antioxidant, antitumor, and anticancer agents [24,25]. Rosmarinic acid (RA), a bioactive phenolic compound, was first isolated from *Rosmarinus officinalis* L. [26] and also found in numerous other plants classified in the subfamily *Nepetoideae* and family *Lamiaceae* [27,28]. Many studies have proved that RA has multiple biological activities, such as anti-inflammatory, antiviral, antibacterial, antioxidant, anti-cancer and so on [29,30,31,32]. RA significantly decreases IL-1β and TNF-α release and ameliorates collagen-induced arthritis in vivo [33,34]. RA showed high antioxidant activities in vivo by scavenging free radical and reactive oxygen species [35]. RA was proved to have antibacterial properties against *Escherichia coli*, *Staphylococcus aureus*, *Salmonella* and *Bacillus subtilis* by destroying bacterial cells and inhibiting the activity of Na+/K+-ATPase in cells [36,37,38]. In recent years, RA has attracted much attention because of its possible application in the medical and nutritional fields [39,40]. Fortunately, RA has catechol groups which are the same as dopamine. Based on the structure and bioactive activities of RA, we assumed that hydrogel grafted with RA would not only have strong tissue adhesion, but also have a variety of activities to promote wound healing.

In this study, RA-grafted dextran/gelatin hydrogel was designed and characterized systemically. Firstly, dextran was oxidized to obtain aldehyde groups, and RA was grafted onto gelatin through a cross-linking reaction mediated by N-(3-Dimethylaminopropyl)-N’-ethylcarbodiimide hydrochloride (EDC·HCl) and N-Hydroxysuccinimide (NHS). Then the oxidized dextran (ODex) and RA-modified gelatin (AG−RA) soon formed ODex−AG−RA hydrogel based on the Schiff base reaction. The morphology, swelling and degradation properties, rheological properties and adhesion properties of ODex−AG−RA hydrogels were characterized. The biocompatibility of ODex−AG−RA hydrogel was verified by cell and blood compatibility experiments. Next, the antibacterial and antioxidant activities of ODex−AG−RA hydrogel were evaluated in vitro. Finally, skin wound healing experiments in rats were carried out to evaluate the efficacy of ODex−AG−RA hydrogel in promoting skin wound healing.

## 2. Results and Discussion

### 2.1. Design and Structural Characterization of the Hydrogels

In recent years, hydrogel, a wet composite material, has attracted much attention. It has become one of the most promising materials for wound healing due to its ideal properties, such as biomimetic 3D structure and moist environment. Dextran and gelatin are usually used to prepare medical hydrogels for their high biocompatibility, hydrophilicity and biodegradability. Additionally, dextran and gelatin have inherent bioactive effects and could acquire multiple functions from a rational design and modified procedure [41]. RA is a common phenolic compound found in more than 30 families of plants and has many remarkable biological activities, such as anti-inflammatory, antioxidant, antiviral and antibacterial activities. These characteristics of RA have led it to be an interesting natural molecule in biomedical applications. Many kinds of medical materials containing RA have been studied, such as polygalacturonic acid/RA membranes and chitosan–RA conjugates [42,43]. RA has a catechol structure similar to dopamine, which could enhance the tissue adhesiveness of hydrogel. Based on the catechol structure being similar to dopamine and the bioactive activities of RA, we designed a series of dynamic cross-linked ODex−AG−RA hydrogels. Grafting RA onto AG would endow ODex−AG−RA hydrogel with strong tissue adhesiveness and strong anti-inflammatory, antioxidant and antibacterial activities.

As shown in Figure 1a, two vicinal hydroxyl groups (-OH) of dextran were oxidized to aldehyde groups (-CHO) with sodium periodate to increase the reactivity of dextran. Fourier transform infrared spectroscopy (FT-IR), hydrogen nuclear magnetic resonance spectroscopy (^1^H-NMR) and UV-vis were used to characterize the structure of ODex. The results showed that the maximum absorption of ODex occurred at 238 nm on the UV spectrum (Appendix A), and the aldehyde group characteristic peak appeared at 1733 cm^−1^ on the FT-IR spectrum (Appendix A). The absorption peak attributed to the aldehyde group proton also appeared at 9.26 and 8.46 ppm, and a new wide peak attributed to the hemiuronic group appeared between 4.2–6.0 ppm on the ^1^H-NMR spectrum (Appendix A). The above results proved the synthesis of ODex. The oxidation degree of ODex determined by hydroxylamine hydrochloride titration was 58.44 ± 2.92%. Meanwhile, aminated gelatin (AG) was synthesized by amination of carboxyl groups on the gelatin chain. Then AG−RA was obtained by grafting RA onto AG mediated by EDC/NHS. The chemical structure of AG and AG−RA were also characterized by FT-IR, ^1^H-NMR, UV-vis and differential scanning calorimetry analysis (DSC). The absorption peak at 325 nm on the UV spectrum (Appendix A) and the appearance of the benzene ring proton peak at 6.3–7.7 ppm in the ^1^H-NMR spectrum (Appendix A) demonstrated the grafting of RA. The content of amino groups was determined by ninhydrin colorimetry [44]. The results showed that the amino group content in the AG was 0.56 ± 0.02 mmoL/g, which was significantly higher than that of gelatin (0.18 mmoL/g). The RA content in the AG−RA was 0.093 ± 0.004 mmoL/g. In addition, the synthesis of AG and AG−RA was also confirmed by DSC (Appendix A). The above results indicated that ODex, AG and AG−RA were successfully prepared. The hydrogel prepolymer solution was obtained by mixing the ODex and AG−RA solutions. The schematic diagram of hydrogel formation is shown in Figure 1b. The -CHO in ODex can spontaneously react with the amino group (-NH_2_) in AG−RA through the Schiff base reaction to form a cross-linking network. As shown in Figure 1c, the solution changed from a flowing state to a columnar hydrogel after mixing ODex solution and AG−RA solution. Three kinds of ODex−AG−RA hydrogels were obtained by adjusting the volume ratio of ODex/AG−RA from 0.5 to 2 (Figure 1d). The ODex−AG hydrogel with a volume ratio of 1 was chosen as a control group. The chemical structures of ODex−AG−RA and ODex−AG hydrogel were confirmed by FT-IR (Figure 1e). The new peak at 1654 cm^−1^ in the FT-IR spectrum was assigned to C = N of hydrogel, which confirmed the formation of a Schiff base bond.

### 2.2. Physical Properties of the Hydrogels

The gelation time of hydrogel is crucial in clinical and biomedical applications. As shown in Figure 2b, the gelation times of the four hydrogels were all less than 200 s. Compared with ODex−AG−1, the gelation time of ODex−AG−RA was different, which indicated that RA was involved in the gelation process. Among the three groups of ODex−AG−RA hydrogels, the gelation time first significantly decreased and then slowly increased with the increase of the AG−RA ratio. The gelation rate of ODex−AG−RA−1 was the fastest due to rapid formation of imine and hydrogen bonds. The ideal gelation time of in situ gel-formation hydrogel dressing is approximately 50–60 s [45], and the gelation time of ODex−AG−RA−1 hydrogels was 61.6 ± 2.8 s, which could meet the requirements of in situ gel-forming hydrogel in vivo.

The three-dimensional (3D) porous network structure of hydrogels plays an important role in the practical application of hydrogel as a channel for the transport of water vapor and other nutrients into the hydrogel. As shown in Figure 2c, all the hydrogels exhibited a porous structure, and the pore size distribution was relatively uniform (Figure 2d). Compared with ODex−AG−1, ODex−AG−RA hydrogels had smaller pore size, which demonstrated that RA participated in the formation of hydrogel and affected the three-dimensional structure. The mechanism was that the phenolic hydroxyl and carbonyl groups in RA formed hydrogen bonds in the hydrogel, and the increased hydrogen bond made the structure denser. Moreover, the pore size of ODex−AG−RA−1 was smaller than the other two groups of ODex−AG−RA hydrogels. The result indicated that the cross-linking density of ODex−AG−RA−1 was highest. Smaller pore size endowed the material with larger specific surface area, which was conducive to the supply of oxygen and nutrients and the removal of waste.

The three-dimensional network structure and swelling property of hydrogels facilitate the penetration of nutrients and the growth of cells inside. An appropriate swelling rate is helpful to maintain a moist wound environment and absorb tissue exudate. Therefore, the swelling performances of the prepared hydrogels were tested. The swelling curves of a series of hydrogels in PBS are shown in Figure 3a. The cross-linking density of the hydrogels has a strong influence on swelling rate. A dense network structure with small pores could hinder the inward diffusion of water. Thus, the swelling rate of ODex−AG−1 was highest because of its largest pore size. The results were consistent with the results of observation with a scanning electron microscope (SEM). Moreover, the swelling rates of ODex−AG−RA−0.5 (497.73%), ODex−AG−RA−1 (233.60%) and ODex−AG−RA−2 (203.79%) were all higher than 200% and decreased with an increase in the content of ODex. The result suggested that the ODex−AG−RA hydrogel could absorb a large quantity of water to maintain a moist environment near a wound.

Degradation capacity is an essential characteristic of tissue engineering materials. Wound dressings should have a desirable degradation rate and the by-products after degradation should be non-toxic. The degradation behavior of these hydrogels was evaluated by simulating the physiological microenvironment. As shown in Figure 3b, the degradation curves of four groups of hydrogels were measured in PBS (pH 7.4) at 37 °C. It can be seen from Figure 3b that differing ratios (cross-linking degrees) have a definite impact on the degradation of hydrogels in vitro. We found that all the hydrogels underwent relatively rapid initial degradation in the first week, and their weights gradually decreased over time. Obviously, the degradation rate of ODex−AG−RA−1 hydrogel was the slowest due to its enhanced cross-linking density. After 28 days, the residual weight of ODex−AG−1, ODex−AG−RA−0.5, ODex−AG−RA−1 and ODex−AG−RA−2 remained at about 74%, 74%, 79%, and 64% respectively, demonstrating that ODex−AG−RA hydrogels showed tunable degradation rate.

Compared with ODex−AG−1, ODex−AG−RA hydrogels had a shorter gelation time, smaller pore structure, less swelling and a slower degradation rate. The above results showed that RA participated in the formation of the three-dimensional structure of the hydrogel, resulting in the formation of hydrogen bonds among a large number of catechol groups, hydroxyl groups and amino groups in the system. Thus, the cross-linking sites increased and a closer cross-linking network was formed. At the same time, the performance of hydrogels can be affected by different mixing ratios of ODex and AG−RA.

### 2.3. Rheological Properties

The formation of hydrogels was monitored by dynamic rheometer. The storage modulus (G′) and loss modulus (G″) of the hydrogels versus time were detected with a rheometer. As shown in Figure 3c, at the beginning, G″ was higher than G′, indicating that the solution was still in the flow state and the hydrogels showed liquid properties. As the Schiff base reaction went on, G′ and G″ gradually increased over time and at some point, G′ and G″ intersected, which was the gelation point of the hydrogels. The time needed to form the hydrogels was consistent with the previous results from the tube inversion method. After that, G′ continued to increase and was consistently higher than G″, indicating that the elastic modulus of the hydrogels gradually increased and the hydrogels behaved as viscoelastic solids. The G′ value of ODex−AG−1 was lower than 10^4^ Pa, which was similar to the reported values of oxidized dextran–gelatin hydrogel or collagen/oxidized dextran hydrogel [46,47]. The G′ value of ODex−AG−RA−1 was greater than ODex−AG−1. With their higher cross-linking density, the G′ of ODex−AG−RA−1 hydrogels was 1.31 × 10^4^ Pa, which was larger than that of the skin (G′ = 200~2000 Pa) [45]. Thus, the ability to resist external force damage of ODex−AG−RA−1 hydrogel was greater than that of skin.

### 2.4. Adhesive Property

Effective adhesion is essential for wound dressings, allowing the dressings to adhere tightly to the tissue, resist shedding and prevent infection. Therefore, the tissue adhesion of hydrogels was evaluated by lap shear testing. As shown in Figure 3d, the adhesion strength of ODex−AG−1 ODex−AG−RA−0.5, ODex−AG−RA−1 and ODex−AG−RA−2 were 23.12 ± 0.38 kPa, 21.97 ± 2.64 kPa, 27.30 ± 2.02 kPa, 23.07 ± 2.30 kPa, respectively. The result demonstrated that four kinds of hydrogels all had strong adhesive ability, which could enable each hydrogel to adhere effectively to the surface of wound. Additionally, compared with ODex−AG−1 or the other two groups of ODex−AG−RA hydrogel, ODex−AG−RA−1 had the highest adhesive strength. The excellent tissue adhesion of ODex−AG−RA−1 was mainly due to the strong covalent interaction and hydrogen bond interaction between the -CHO, the catechol groups in the hydrogels and the amino and mercaptan groups in the tissue, which improved their adhesion performance [48]. The results indicated that the grafting of RA could effectively improve the adhesion of the hydrogel, and the adhesion strength of the hydrogel could be adjusted according to the mixing ratio.

### 2.5. In Vitro Antioxidant and Antibacterial Activities

An excessive number of free radicals and reactive oxygen species (ROS) around the wound site might result in oxidative stress [49], which could slow wound healing and even lead to more serious inflammation. Bioactive hydrogels with the ability to clear ROS might promote wound healing. Therefore, the development of wound dressings with reductive or antioxidant activities is of great significance. RA is a nature-derived antioxidant factor. The excellent antioxidant activities of RA are mainly attributed to the two catechol groups in its molecular structure. By introducing RA into the hydrogel system, the ODex−AG−RA hydrogels were expected to show a strong antioxidant effect in the treatment of chronic wounds. The antioxidant ability of each hydrogel was evaluated by determining their 1, 1-diphenyl-2-picrylhydrazyl (DPPH) free radical scavenging, superoxide anion radical scavenging and total reductive powers. The results are shown in Figure 4a. ODex−AG−RA hydrogels all showed higher DPPH free radical scavenging ability (>95%) than ODex−AG−1 hydrogel (74.70%). Meanwhile, the superoxide anion radical scavenging rate of ODex−AG−RA−1 was 1.64 times that of ODex−AG−1. Moreover, ODex−AG−RA hydrogels also showed an obvious increase in total reductive properties compared with ODex−AG−1. All the results demonstrated that ODex−AG−RA hydrogels had higher free radical scavenging abilities and reductive powers than ODex−AG−1, which indicates that grafted RA could improve the antioxidant activity of hydrogels. The results were consistent with a published study about chitosan-RA conjugates [43]. Overall, the excellent free radical scavenging ability of ODex−AG−RA hydrogels make it a candidate for clinical wound dressing.

RA has also been reported to have a broad spectrum of antimicrobial capabilities and to inhibit the proliferation of Gram-positive and Gram-negative bacteria. Due to this advantage of RA, ODex−AG−RA hydrogels also exhibit the ability to inhibit bacterial proliferation. 100 uL of hydrogels were co-cultured with *E. coli* or *S. aureus* for 18 h, and their antibacterial ability was evaluated by the colony-formation counting method. As shown in Figure 4b,c, the number of colonies after co-culturing with hydrogels was obviously lower than the control group. All the groups treated by the hydrogels exhibited a 100% mortality rate for *S. aureus*. The bactericidal efficiency rate of the ODex−AG−1 group against *E. coli* was 89.14 ± 3.71%. Meanwhile, the bactericidal efficiency rates of ODex−AG−RA−0.5, ODex−AG−RA−1 and ODex−AG−RA−2 against *E. coli* were 94.76 ± 2.65%, 97.00 ± 2.12%, and 89.70 ± 10.33%, respectively, which were higher than that of ODex−AG−1 group. The antibacterial properties of the hydrogel were further verified by live and dead staining (Figure 4d). The control group showed green fluorescence with some red fluorescence, while the hydrogel groups showed more red fluorescence. This indicated that more bacteria died after co-culturing with hydrogels. Additionally, the red fluorescence in ODex−AG−RA−1 group was greatest in both *E. coli* and *S. aureus*. These results proved that the hydrogels grafted with RA have stronger antibacterial properties, especially ODex−AG−RA−1. The above results also confirmed the excellent in vitro antibacterial property of ODex−AG−RA hydrogels against both Gram-positive and Gram-negative bacterial. 

In order to clarify the antibacterial mechanism of the prepared hydrogels, the SEM images of *S. aureus* and *E. coli* on the hydrogels after co-culturing for 18 h were taken and shown in Figure 4e. The structure of the cytoderm of *E. coli* and *S. aureus* in the PBS control group was complete and smooth. However, after co-culturing with hydrogels, the wall of the bacterial cell was found to have collapsed with the intracellular fluid leaking out. The incomplete membrane structure affected normal metabolism and resulted in the death of the bacteria. In addition, the antibacterial mechanism of the hydrogel was also verified by a formation test of bacterial biofilm. As shown in Figure 4f,g, the quantitative analysis of the biofilm biomass was carried out by spectrophotometry. The biofilm biomass of the blank group increased significantly with the extension of time, while the rise of biomass in the ODex−AG−RA hydrogel groups was significantly inhibited. Furthermore, the biomass in the ODex−AG−RA hydrogel groups was significantly less than that in ODex−AG−1, especially for ODex−AG−RA−1. Moreover, all samples exhibited slightly higher activity against *S. aureus* than *E. coli*. In conclusion, the above results confirmed that ODex−AG−RA hydrogels can effectively resist the formation of biofilm in vitro.

The above results indicated that ODex−AG−RA hydrogels had stronger antibacterial activity compared with ODex−AG−1. They exerted an antibacterial effect mainly by destroying the bacterial cell wall and inhibiting biofilm formation. The positively charged amino groups in the hydrogels could damage the walls of the bacteria. The inherent antibacterial properties of RA as well as the Schiff bases, including aromatic ring, also played an important role in antimicrobial activity.

### 2.6. Hemocompatibility

High blood compatibility is a prerequisite for the application of biomaterials. The hemocompatibility of ODex−AG−RA hydrogels was evaluated by hemolysis examination in vitro. Figure 5a shows hemolysis after incubation with the samples for 1 h. The supernatant in the hydrogel sample groups and the negative control group were all transparent, whereas the supernatant of the positive control group turned red due to the rupture of red blood cells. Quantitative results showed that the hemolysis ratio of all groups of hydrogels was less than 3%. The hemolysis ratio of ODex−AG−RA−0.5, ODex−AG−RA−1 and ODex−AG−RA−2 hydrogels were 2.94%, 0.05% and 0.50%, respectively, which was lower than the international standard of 5%. These results indicated that the prepared hydrogels all had high blood compatibility.

### 2.7. In Vitro Cell Compatibility

Biocompatibility with fibroblasts is essential for wound dressing, which is usually in direct contact with the skin or other connective tissue. Firstly, the cytotoxicity of ODex−AG−RA hydrogels was tested by the leaching solution method. As shown in Figure 5b, the viability of all L929 cells treated with a different leaching solution from each hydrogel was greater than 85% compared with the control group, indicating that the cells grew well during the whole experiment. In addition, there was no significant difference in cell viability between different concentrations of hydrogel leaching solution, which showed these hydrogels had high cell compatibility and the hydrogel leaching solution had almost no cytotoxicity. Meanwhile, the Live/Dead staining assay (Figure 5c) showed that almost all L929 cells were green and spindle-shaped, and there were few dead cells (stained in red). This result was consistent with the quantitative results of CCK-8, indicating high cytocompatibility of the prepared hydrogels.

### 2.8. In Vivo Wound Healing and Histological Evaluation

The full-thickness skin incision defect of male SD rats was used as a model to study wound repair with the intervention of hydrogels (Figure 6a). According to the above experimental results, ODex−AG−RA−1 hydrogel had high mechanical strength, tissue adhesion and antibacterial and antioxidant effects. Therefore, ODex−AG−RA−1 was selected as the experimental group to study the promotive effect of ODex−AG−RA hydrogel on wound healing, while ODex−AG−1 was selected as the control group. The wounds without a dressing treatment were used as a blank control group, and the wounds treated with 3M Tegaderm^TM^ (a commercial dressing) were used as a positive control group.

#### 2.8.1. Macroscopic Observation and Histopathology

Images of wounds at different points in the healing process are shown in Figure 6b, and the degree of wound healing was quantitatively analyzed (Figure 6c–e). In the early stage of wound healing, the wound contraction of the hydrogel groups was significantly better than that of the blank control group and Tegaderm group. On day 3, the wound covered with the Tegaderm^TM^ hydrogel showed redness, while the ODex−AG−RA−1 hydrogel groups showed no such signs. On day 7, the wound healing rates of all groups were over 60%. The wound healing rates of the blank control group and Tegaderm group were 65.91 ± 4.88% and 69.79 ± 6.03%, respectively. The wound healing rates of the ODex−AG−1 and ODex−AG−RA−1 hydrogel groups were 86.26 ± 2.98% and 87.15 ± 3.30%, respectively. The wound closure rate of the ODex−AG−RA−1 group was about 21.24% higher than that of the blank group. On day 14, the wound area in all groups was significantly reduced, and the strongest treatment effect was observed in the ODex−AG−RA−1 hydrogel group. On day 21, the wound contraction rate of the control group and the Tegaderm group were 95.26% and 96.84% respectively, while the wounds in the ODex−AG−RA−1 hydrogel groups were almost completely closed and covered with hair. These results showed that ODex−AG−RA−1 hydrogel with RA grafting had a stronger ability to promote wound healing than 3M Tegaderm^TM^ and ODex−AG−1. This might be attributed to the biological reductive, antibacterial and antioxidant properties of RA, which could accelerate the wound repair process and thus exhibit superior effects in wound healing.

The wound tissues were collected on the 3rd, 7th, 14th and 21st days, respectively. Histomorphological analysis of skin sections was performed by H&E staining and Masson’s trichrome staining. As demonstrated in Figure 7a, mild inflammatory reaction was observed in all groups on day 3, and a large number of inflammatory cells gathered in the wound site. Only a small number of inflammatory cells migrated to the injury site in the ODex−AG−RA−1 group, which may be attributed to its ideal antioxidant antimicrobial and anti-inflammatory properties. On day 7, some purple markers of inflammatory cell infiltration could still be observed in the blank control group, Tegaderm group and ODex−AG−1 group. The inflammation in the ODex−AG−RA−1 group had obviously receded, and a small amount of epithelial tissue and hair follicles had formed in this group. The results indicated that ODex−AG−RA−1 hydrogel could effectively reduce inflammatory cells and inhibit acute and chronic inflammation in wounds, which was also observed in the study of polygalacturonic acid/RA membranes [42]. On day 14, a large amount of granulation tissue was observed in the regeneration area of the control group and the Tegaderm group, while only a small number of blood vessels and hair follicles were generated. Most of the granulation tissue in the ODex−AG−RA−1 group was replaced by well-arranged collagen fibers and blood vessels. On day 21, a large number of blood vessels and hair follicle structures were generated in the ODex−AG−RA−1 group, and the wound structure was similar to normal rat skin tissue. The wound regeneration tissue in the other three groups was also basically mature, but the healing was significantly slower than in the ODex−AG−RA−1 group.

Masson staining results are shown in Figure 7b. A light blue color was observed in the Masson staining diagram on day 7, which demonstrated that the overall amount of collagen deposition at the wound site was low by the 7th day of wound healing. This was because relatively few fibroblasts infiltrated the wound site within 7 days after surgery. On day 14, the wound was in the proliferative and remodeling stage. During this period, the number of fibroblasts increased, and the synthesis and deposition of collagen were significantly enhanced. The Masson staining pictures showed a darker blue color, and the color of the ODex−AG−RA−1 group was significantly deeper. Then we quantitatively characterized collagen deposition at the wound site by measuring the intensity of the blue stained area. As shown in Figure 7c, the amount of collagen deposition on wounds in ODex−AG−RA−1 group was 6.3 times that of the control group on day 7. On day 14, the collagen deposition amount in ODex−AG−RA−1 group was 4.31 times and 1.40 times of that in the control group and ODex−AG−1 group, respectively. These results indicate that ODex−AG−RA−1 hydrogel could accelerate wound healing by promoting collagen synthesis and deposition in wound tissue during the proliferative stage of wound healing.

#### 2.8.2. Content of Antioxidant Factor in Regenerated Skin Tissue

The aggregation of activated macrophages after tissue injury can trigger oxidative stress. MDA is a classic biomarker of oxidative stress and its content reflects the level of oxidative stress in the body [50]. H_2_O_2_ is a two-electron oxidizing agent producing both free and non-free radical substances, which can damage various cells [51]. A high content of MDA and H_2_O_2_ in the wound could delay healing by inducing an inflammatory reaction and apoptosis. Dressings with antioxidative activity have been proved to promote wound healing. The content of H_2_O_2_ and MDA during wound healing was measured to evaluate the antioxidant effect of hydrogels (Figure 7d,e). The quantitative results of H_2_O_2_ and MDA showed that MDA content in the control group was 1.21 times that in the ODex−AG−RA−1 group on the 7th day after injury. H_2_O_2_ content in the control group was 1.23 and 2.38 times that in the ODex−AG−1 and ODex−AG−RA−1 groups, respectively. Compared with the other groups, the ODex−AG−RA−1 hydrogel group showed the lowest content H_2_O_2_ and MDA. These results indicated that the ODex−AG−RA−1 hydrogel had definite antioxidative activity and could obviously decrease the oxidative stress level at the wound site. The result was consistent with antioxidant experiments in vitro. The excellent antioxidant ability of ODex−AG−RA−1 in vitro and in vivo must be related to RA, which is a natural antioxidant compound. In addition, the physical barrier formed by the hydrogels prevented the production of reactive oxygen species caused by exogenous microbial infections.

#### 2.8.3. Expression of Inflammatory Factors

Macrophages play an important role in the process of wound healing. Macrophages are divided into three types: the M0 type is inactive, the M1 type releases proinflammatory cytokines and the M2 type is anti-inflammatory and can promote collagen deposition and repair. M1 and M2 macrophages can transform into each other at different stages of wound healing, which helps the wound transfer smoothly from the inflammatory stage to the proliferative stage. TNF-α is a typical proinflammatory cytokine which is highly expressed in the inflammatory response [52,53]. CD163, a typical surface marker of M2 macrophages, plays an important role in the regulation of inflammation [54]. The expressions of TNF-α and CD163 during wound healing were measured by immunohistochemistry to evaluate inflammatory response in this study. As demonstrated in Figure 8a,d, the expression of TNF-α in the ODex−AG−RA−1 group was significantly lower (*p* < 0.05) than in the other groups on day 7 and day 14. The result proved that ODex−AG−RA−1 hydrogel could inhibit the inflammation during wound healing. Moreover, compared with day 7, the TNF-α content in all groups decreased on day 14. The TNF-α content in the control group was 5.28 times that of the ODex−AG−RA−1 group on day 14. This indicated that the hydrogel group accelerated the process of inflammatory reaction by reducing the expression of pro-inflammatory factor TNF-α. Conversely, the expression of CD163 in the ODex−AG−RA−1 group was highest on day 7, which indicated the number of M2-phenotype macrophages near the wound was higher in the ODex−AG−RA−1 hydrogel group. The result indicated ODex−AG−RA−1 could promote the transformation of M1 macrophages into M2 macrophages, and thus the wound could transition from the inflammatory stage to the proliferative stage. Compared with day 7, the expression of CD163 decreased on day 14 except in the control group. The expression of CD163 in the ODex−AG−RA−1 group was lowest. On day 14, the relatively high expression of TNF-α and CD163 in the control group suggested that it was still in the chronic inflammatory stage, while theODex−AG−RA−1 group’s inflammatory reaction process was accelerated and transitioned to the proliferative stage of wound healing successfully and with the lowest expression of TNF-α and CD163. All these results proved that hydrogel-grafted RA could shorten the inflammatory period and thus accelerate wound healing.

#### 2.8.4. Neovascularization in Regenerative Skin Tissue

Neovascularization was evaluated by immunofluorescence staining of the vascular endothelial specific marker CD31 during the proliferative and remodeling stages of wound healing [55]. As shown in Figure 8c,f, significant red fluorescence was observed in the ODex−AG−RA−1 group, showing more CD31 expression. The relative coverage area was 1.28, 1.66, and 2.33 times that of the ODex−AG−1, Tegaderm and control groups, respectively. The result indicated that ODex−AG−RA−1 could promote neovascularization; thus, wound healing would be promoted and the quality of regenerated skin was also improved. This might be related to the anti-inflammatory and antioxidant activities of ODex−AG−RA−1. Additionally, dextran and gelatin are natural polymers, which could provide nutrients for tissue regeneration and be conducive to cell growth and proliferation.

## 3. Materials and Methods

### 3.1. Materials

Dextran (MW = 40 kDa), sodium periodate (NaIO_4_), Ethylenediamine hydrochloride(ED·HCl), EDC·HCl, NHS and RA were obtained from Aladdin Biological Technology Co., Ltd. (Shanghai, China). Gelatin (MW = 60 kDa) from fish skin was obtained from Sigma–Aldrich; Cell Counting Kit-8 (CCK-8) and Calcein/PI Cell Viability/Cytotoxicity Assay Kit were bought from Beyotime Biotechnology (Shanghai, China). All other reagents used in the experiment were analytical grade without further purification.

### 3.2. Preparation of ODex

Dextran was oxidized into ODex by NaIO_4_ according to the reported method [56]. Briefly, dextran was stirred and dissolved in 50 mL deionized water, and NaIO_4_ solution was added to the dextran solution in the ice bath. The reaction was allowed to proceed at 25 °C with stirring for 4 h. Finally, diethylene glycol was added to stop the reaction. ODex was obtained after dialysis and freeze drying. The oxidation degree of ODex was confirmed by hydroxylamine hydrochloride titration [57]. The structure of ODex was characterized by FT-IR, ^1^H-NMR and UV-vis. The specific processes are shown in the Appendix A.

### 3.3. Preparation of AG and AG−RA

Gelatin was aminated with ED·HCl to obtain AG according to our previous study [58]. Briefly, 5 g gelatin was dissolved in 200 mL NaH_2_PO_4_ (0.1 moL/L, pH 5.0) buffer solution. ED·HCl was added into the clear solution. The reaction mixture was stirred at 37 °C for 6 h under catalysis. AG was obtained after dialysis and freeze drying. Then, AG−RA was synthesized by grafting RA onto AG mediated by EDC/NHS. First, RA was dissolved in ethanol and activated by adding EDC and NHS in an ice bath for 1 h. Then the mixture was dropped into AG solution and stirred for 12 h at room temperature. Finally, the reaction solution was dialyzed and freeze-dried successively to obtain AG-RA. The structure of AG and AG−RA was characterized by FT-IR, ^1^H-NMR, UV-vis and differential scanning calorimetry (DSC). The amino group content in the AG was determined by Ninhydrin colorimetry. The grafting rate of RA was determined by UV-vis. The specific processes are shown in the Appendix A.

### 3.4. Preparation of ODex−AG−RA Hydrogel

First, ODex and AG−RA were dissolved in PBS (pH7.4) at a concentration of 0.15 g/mL and 0.30 g/mL, respectively. Then ODex and AG−RA were mixed in the tube and stirred homogenously at ratios of 0.5:1, 1:1 and 2:1. After full stirring, the mixed solution was observed carefully to record the gelation time. For the convenience of subsequent statements, the hydrogels prepared by ODex and AG−RA at different ratios are abbreviated as ODex−AG−RA−0.5, ODex−AG−RA−1 and ODex−AG−RA−2. In addition, the hydrogel based on ODex and AG−RA at a ratio of 1:1 was also prepared as the control sample and abbreviated as ODex−AG−1. FT-IR were used to confirm the successful preparation of hydrogels.

### 3.5. Characterization of Physical and Chemical Properties

#### 3.5.1. Gelation Time

The gelation time was determined by vial inversion and recorded when the mixture solution stopped flowing. The specific processes are shown in the Appendix A.

#### 3.5.2. Morphology Analysis

The internal morphology of hydrogels was determined using a SEM after freeze drying. The specific processes are shown in the Appendix A.

#### 3.5.3. Swelling and In Vitro Degradation Test

The hydrogels were immersed in PBS (pH7.4) at 37 °C to determine their swelling performance in vitro. To determine their degradation properties, the hydrogels were immersed in PBS (pH 7.4) and their degradation rate was determined by the weight loss method [59]. The specific processes are shown in the Appendix A.

#### 3.5.4. Rheological Property of Hydrogels

The modulus of these hydrogels was measured by a TA rheometer (DHR-1), and the modulus changes over time were recorded by oscillation mode–time scanning. The specific process is shown in the Appendix A.

#### 3.5.5. Adhesive Strength Test

The adhesive properties of the hydrogels were tested on porcine skin surfaces through a lap shear test [60]. The specific process is shown in the Appendix A.

### 3.6. Antioxidant and Antibacterial Activities of Hydrogels In Vitro

The antioxidant activity of the hydrogels in vitro was analyzed by DPPH free radical scavenging, superoxide anion radical scavenging and total reducing power [61,62,63].

The antibacterial activity of ODex−AG−RA−0.5, ODex−AG−RA−1, ODex−AG−RA−2 and ODex−AG−1 hydrogel was evaluated by a contact antibacterial test against *E. coli* (Gram-negative bacteria) and *S. aureus* (Gram-positive bacteria). Specifically, a 10^6^ CFU mL^−1^ bacterial suspension was incubated for 24 h with the hydrogels. The co-culture solution was spread and cultured on the agar plate, and the colony forming unit was calculated to evaluate the antibacterial performance. Then the bacterial dead/live staining and morphology analysis were carried out. On this basis, the anti-biofilm properties of hydrogels were studied [64]. The bacterial solution was added to the surface of the hydrogel block and cultured for 24, 48 and 72 h respectively. The biofilm biomass was determined by spectrophotometry after crystal violet staining.

### 3.7. Biocompatibility of Hydrogels

#### 3.7.1. Hemolytic Test

The same volumes of hydrogels were mixed with red blood cell suspension and incubated at 37 °C for 1 h. The absorbance at 540 nm of the supernatant after centrifugation was measured to calculate the hemolysis rate of the hydrogels [65]. The specific process is shown in the Appendix A.

#### 3.7.2. Cytocompatibility

In vitro cytotoxicity testing was performed with L929 cells by the leaching solution method. The cytotoxicity of the hydrogels was tested using the Cell Counting Kit-8 (CCK-8) assay and dead/live cell staining experiments [66]. The specific process is shown in the Appendix A.

### 3.8. In Vivo Wound Closure Evaluation

The full-thickness skin defect model was established on male SD rats (about 250 g, 6 weeks age). The protocol was approved by the Animal Experiment Ethics Committee of Binzhou Medical University (No. BY2021-387). Briefly, four full-thickness skin wounds with a diameter of 13 mm were created on the backs of rats. The four wounds were treated with ODex−AG−RA−1, ODex−AG−1, 3M Tegaderm^TM^ and saline, respectively. For ODex−AG−RA−1 and ODex−AG−1, 50 μL hydrogel was placed on the wound to evaluate its healing effect as a wound dressing. Photographs of the wounds were taken at different time points (3rd, 7th, 14th and 21st day) of healing. The fresh wound tissue of the rats was collected, fixed with neutral formaldehyde, embedded in paraffin and then sectioned for histological examination. The wound contraction rate, histopathologic examination, Masson’s trichrome staining and immunohistochemical and immunofluorescence staining were used to study the wound healing, inflammation reaction, collagen deposition, and revascularization properties of the hydrogels. Further, the antioxidant properties of hydrogels in vivo were evaluated based on the content of malondialdehyde (MDA) and hydrogen peroxide (H_2_O_2_). The experimental details are available in the Appendix A.

### 3.9. Statistical Analysis

All data were presented as mean ± standard deviation. Comparisons in different groups were analyzed by a one-way ANOVA or two-way ANOVA test using SPSS Statistics version 26 (International Business Machines Corp., USA), in which *p* < 0.05 was considered statistical significance.

## 4. Conclusions

In this study, an in situ ODex−AG−RA hydrogel with strong tissue adhesion and multiple bioactive activities was developed as wound dressing. The efficacy of promoting wound healing and the mechanisms of ODex−AG−RA hydrogel were systemically evaluated. Compared with simple ODex−AG−1, the ODex−AG−RA hydrogels had faster gelation time, stronger tissue adhesion, higher antioxidant capacity and antibacterial activity in vitro. The examination of hemolysis and co-culturing with L929 cells showed high in vitro biocompatibility of ODex−AG−RA hydrogels. Moreover, the ODex−AG−RA−1 hydrogel significantly promoted skin wound healing, with more complete skin structure including collagen deposition, hair follicles and blood vessels compared with ODex−AG−1 and 3M Tegaderm^TM^. Further, the mechanism of ODex−AG−RA−1 for promoting wound healing was proved to be related to its anti-inflammatory effect by adjusting the expression of inflammatory cytokines (TNF-α and CD163) and reducing the level of oxidative stress (MDA and H_2_O_2_). In conclusion, this study demonstrated the wound healing efficacy of RA-grafted hydrogels for the first time and provided a strategy for the design of hydrogels for wound dressing. The ODex−AG−RA−1 hydrogel was a promising candidate as a wound dressing.

## Figures and Tables

**Figure 1 molecules-28-04034-f001:**
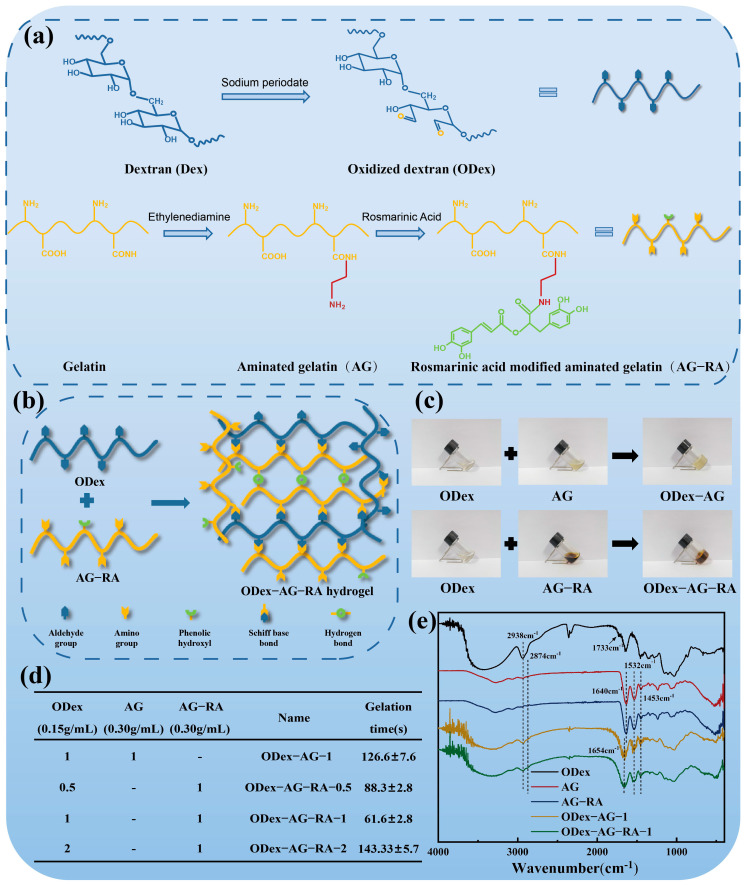
Synthesis and structural characterization of hydrogels. (**a**) Synthesis of ODex, AG and AG−RA. (**b**) Schematic illustration of hydrogel formation. (**c**) Photographs of the ODex, AG, AG−RA solution and ODex−AG, ODex−AG−RA hydrogels. (**d**) Components and gelation time of different hydrogels. (**e**) FT-IR spectra of ODex, AG, AG−RA, ODex−AG and ODex−AG−RA.

**Figure 2 molecules-28-04034-f002:**
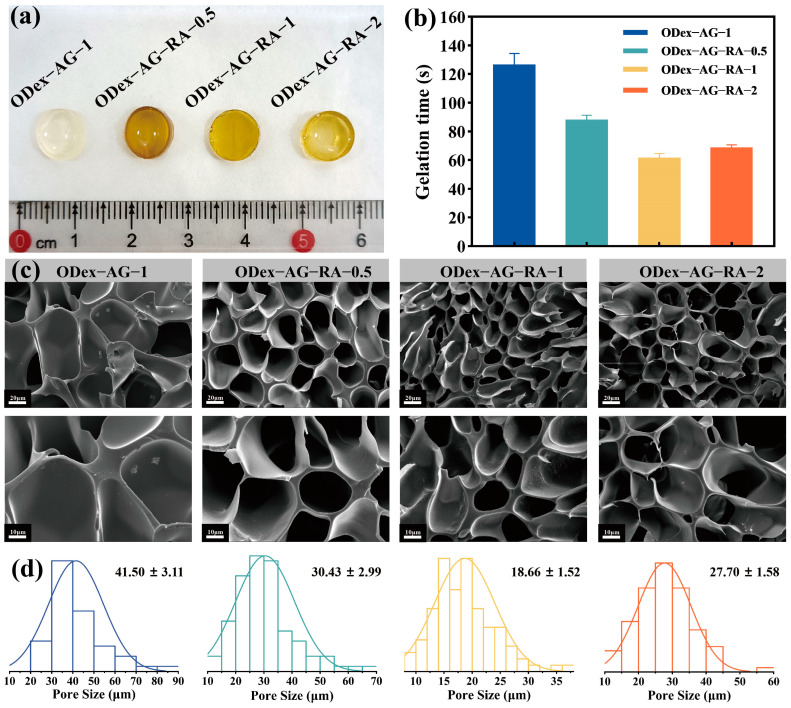
Characterization of gelation time and micro morphology of hydrogels. (**a**) Photographs of different hydrogels. (**b**) Gelation time of different hydrogels. (**c**) Scanning electron microscope (SEM) images of hydrogels. (**d**) Pore size distribution of hydrogels.

**Figure 3 molecules-28-04034-f003:**
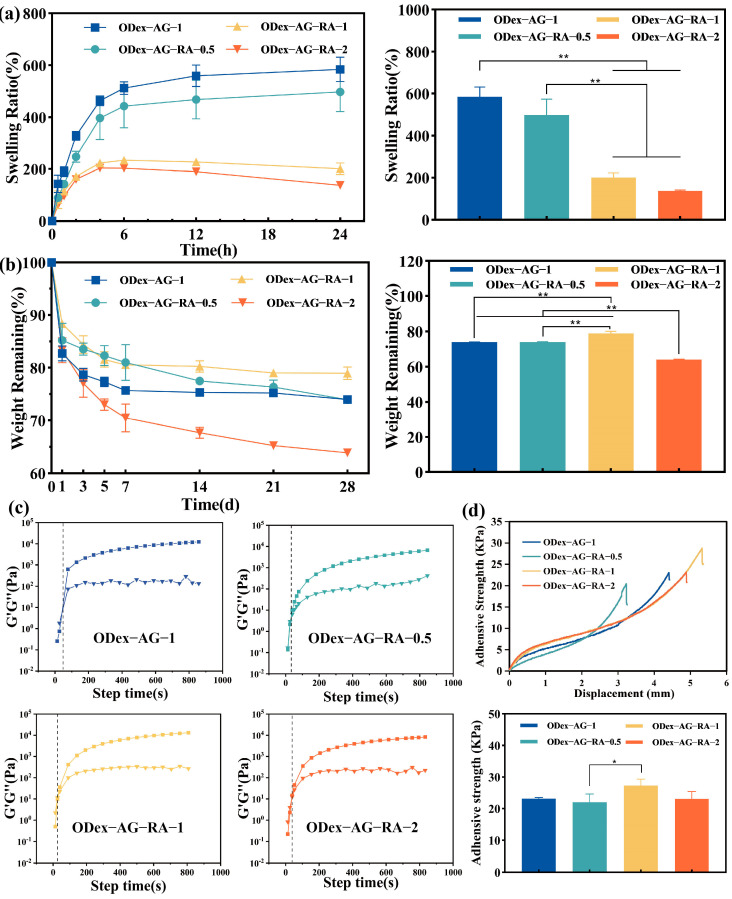
Characterization of hydrogels. (**a**) Equilibrium swelling ratio of hydrogels after swelling for 24 h in PBS (pH 7.4) at 37 °C. (**b**) In vitro degradation curve and weight remaining of hydrogels (*n* = 3); (**c**) Rheological behavior of hydrogels. (**d**) The adhesive strength of hydrogels on porcine skin. * *p* < 0.05, ** *p* < 0.01.

**Figure 4 molecules-28-04034-f004:**
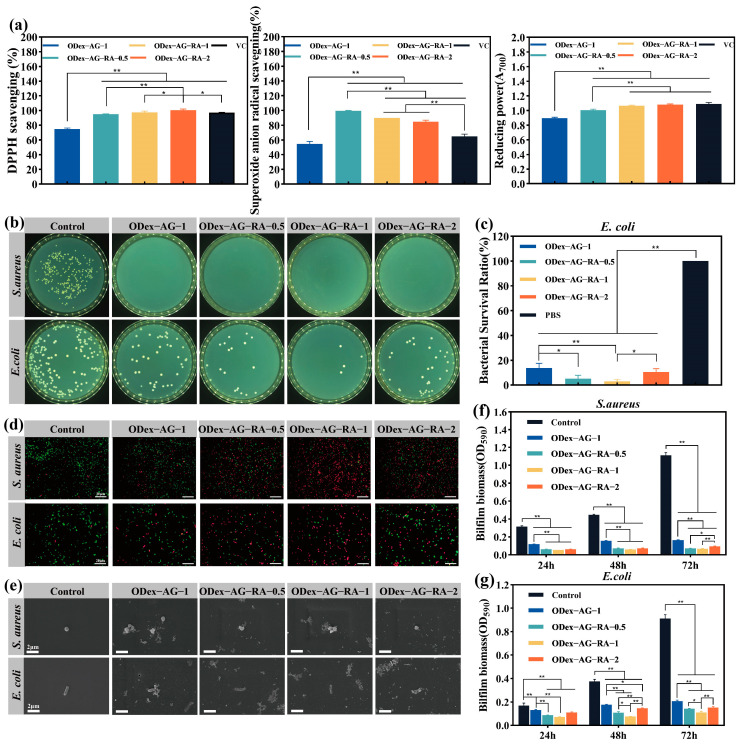
Characterization of antibacterial and antioxidant properties of hydrogels. (**a**) Antioxidant property of hydrogels evaluated by DPPH radical scavenging rate, superoxide anion radical scavenging rate and total reductive power. (**b**) Photographs of survival bacteria clones on agar plates after contacting with hydrogels. (**c**) Quantitative analysis of bacterial survival rate of *E. coli*. (**d**) Live and dead staining of bacteria after co-culture with hydrogels. (**e**) Morphology of bacteria under SEM; biomass of biofilm of *S. aureus* (**f**) and *E. coli* (**g**) after culture for 24 h, 48 h and 72 h with hydrogels. * *p* < 0.05, ** *p* < 0.01.

**Figure 5 molecules-28-04034-f005:**
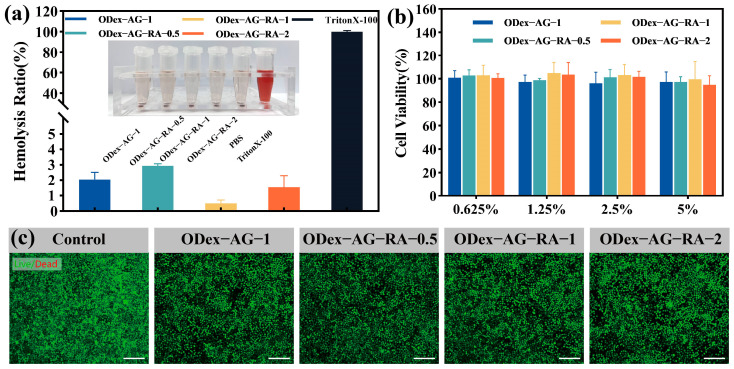
Biocompatibility of hydrogel. (**a**) Hemolysis ratio (%) of hydrogels. (**b**) Cell viability after culturing with hydrogel leaching solution for 24 h; (**c**) LIVE/DEAD staining pictures of L929 cells (*n* = 3). The scale bars in (**c**): 200 μm.

**Figure 6 molecules-28-04034-f006:**
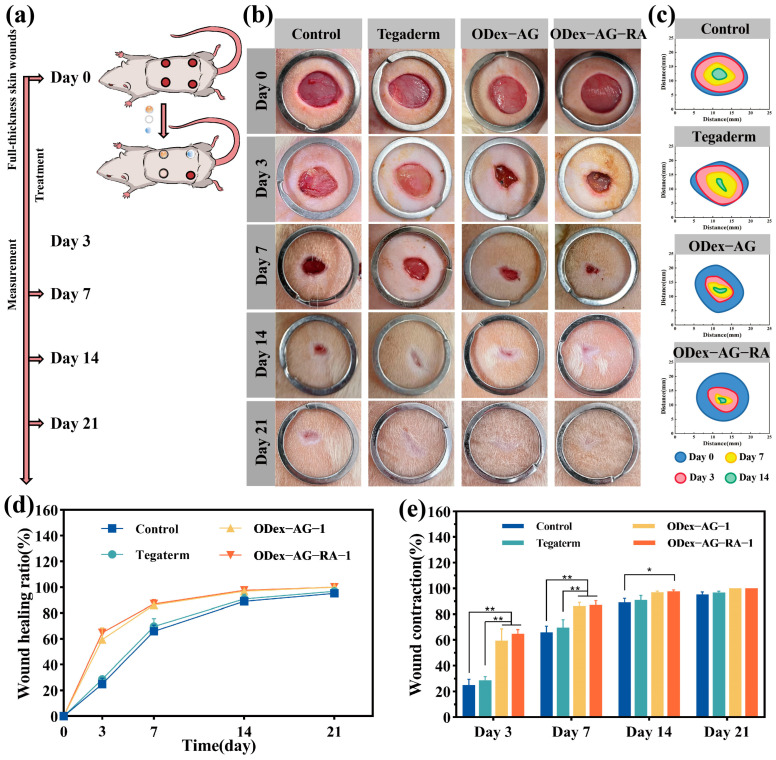
Schematic diagram of rat modeling and wound healing process. (**a**) Schematic illustration of establishment and treatment of full thickness skin defect model in rats. (**b**) Photographs of full-thickness skin wounds on day 0, day 3, day 7, day 14 and day 21. (**c**) Wound areas of different groups at each time point; wound healing rate curves (**d**) and wound contraction (**e**) during wound repair. * *p* < 0.05, ** *p* < 0.01.

**Figure 7 molecules-28-04034-f007:**
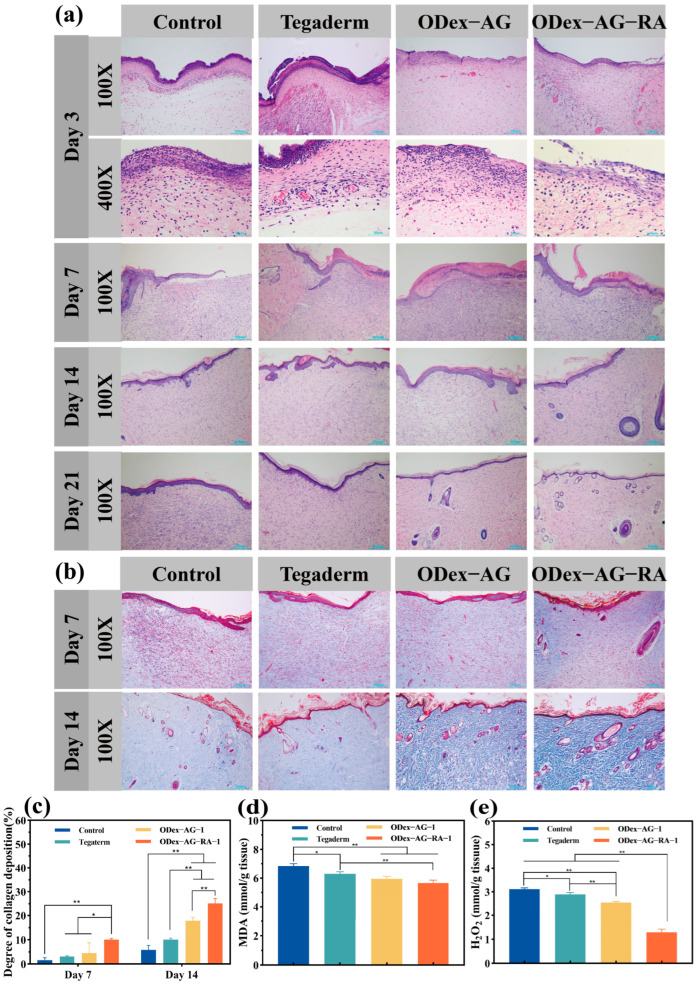
Histological evaluation and analysis of oxidative stress indicators. (**a**) Histomorphological evaluation of the control, Tegaderm, ODex−AG−1 and ODex−AG−RA−1 group. (**b**) Masson staining of regenerated skin wounds for control, Tegaderm, ODex−AG−1 and ODex−AG−RA−1 groups on day 14. (**c**) Quantitative analysis of relative intensity of collagen deposition coverage area for different groups; contents of MDA (**d**) and H_2_O_2_ (**e**) in wound tissue on day 7. * *p* < 0.05, ** *p* < 0.01.

**Figure 8 molecules-28-04034-f008:**
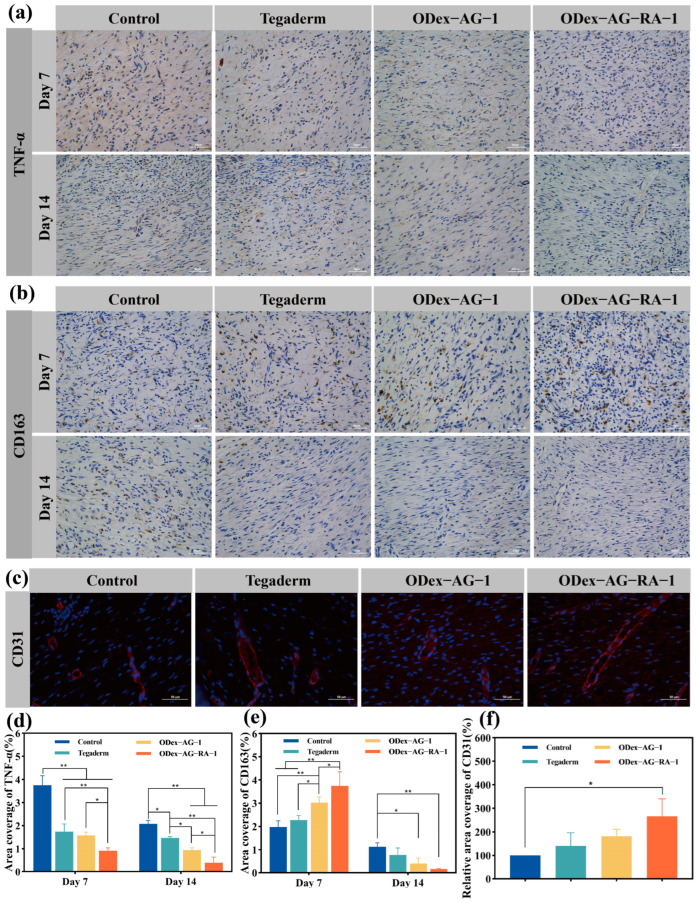
Immunohistochemistry and immunofluorescence analysis. Immunohistochemical staining results of wound regeneration site on day 7 and day 14 with TNF-α (**a**) and CD163 (**b**). (**c**) Immunofluorescence staining of wound regeneration site for control, Tegaderm film, ODex−AG−1 and ODex−AG−RA−1 groups on day 14; quantitative results of TNF-α (**d**), CD163 (**e**) and CD31 (**f**) relative coverage area (*n* = 3). * *p* < 0.05, ** *p* < 0.01.

## Data Availability

All data pertinent to this work are included in the paper.

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
