# Peer review of "Rosmarinic Acid-Grafted Dextran/Gelatin Hydrogel as a Wound Dressing with Improved Properties: Strong Tissue Adhesion, Antibacterial, Antioxidant and Anti-Inflammatory"

_molecules, 2023, doi:10.3390/molecules28104034_

Round 1

Reviewer 1 Report

The manuscript entitled “Rosmarinic acid grafted dextran/gelatin hydrogel as a wound dressing with improved properties: strong tissue adhesion, antibacterial, antioxidant and anti-inflammatory” may be acceptable in Molecules journal after following corrections are made. Recommend for publication after minor revision.

Comment 1: Images can be modified for better clarity.

Comment 2: Reported gel has any effect on other inflammatory pathways.

Comment 3: is there any toxic effect of the gel studied before?

Comment 4: Conclusion part can be better written.

Comment 5: Please correct grammatic and typographic mistakes of manuscript.

Reviewer 2 Report

1.    Abstract is too short and does not clearly express the summary of the work, please correct it. Highlights the significant results with numbers.

2.     Throughout the text, the language must be adequately check. Apart from formal grammar errors, there are many substantial language errors. All the scientific name of Bacteria must be in italic.

3.   The introduction and discussion must be explained more specifical. You need to highlight the mechanism of Rosmarinic acid grafted dextran/gelatin hydrogel as a wound dressing.

In the discussion section you must compare your funding with other studies and presented the mechanism of action of RA grafted dextran/gelatin hydrogel

Please cite these papers and improve this section for instance you need to talk more about the role of natural bioactive compounds.

•     Hepatoprotective effect of nanoniosome loaded Myristica fragrans phenolic compounds in mice-induced hepatotoxicity [DOI: 10.1111/jcmm.17581]

•     Antiproliferation effects of nanophytosome‑loaded phenolic compounds from fruit of Juniperus polycarpos against breast cancer in mice model: synthesis, characterization and therapeutic effects

·       Encapsulated phenolic compounds from Ferula gummosa leaf: A potential phytobiotic against Campylobacter jejuni infection [DOI: 10.1111/jfpp.16802]

4. All the method must be in details (please rewrite). The protocol of synthesized, the encapsulation of RA must determine as well. 

5. All figures presented must be clear. It is very difficult to understand. The quality must be improved. The statistical analysis must be obvious in the data's. 

Reviewer 3 Report

The work is interesting and opens new frontiers for the synthesis, design and application of new hydrogels.

I would suggest providing a more detailed explanation of the hydrogel synthesis and explicitly stating the synthetic steps and structures involved, as the text is difficult to follow. I would also recommend using more immediate and clear abbreviations (e.g. AR and RA are easily confused). I suggest rewriting the text from line 103 to 121 in a clearer and more fluid manner. If references to figures are made in the text, it would be appropriate to include them or let accessible the SI material.

Moreover, authors would justify why ODex-AR-0.5, containing more RA, does not work well as ODex-AR-1, considering a bigger pore size and a better swelling. Why this hydrogel ODex-AR-0.5 has often a behavior more similar to ODex-AG-1 instead of ODex-AD-1.

Several remarks:

Figure 1, a) completely incomprehensible, better to show the synthesis and change the representative model. b) do not add any useful information. c) what means the percentile in ODex(15%), AG(30%) and AR(30%), and is R rosmarinic acid, maybe ratio?

These references must to be added:

ACS Sustainable Chem. Eng. 2014, 2, 1318−1324: doi: 10.1021/sc500154t

J. Mater. Chem. B, 2022, 10, 7148-7160, doi: 10.1039/D2TB00591C

Carbohydrate Polymers 87 (2012) 1749– 1755

Carbohydrate Polymers 273 (2021) 118619.

Round 2

Reviewer 2 Report

The manuscript has been improved and can be publish in the present form.

Reviewer 3 Report

In this revisioned form the manuscript appears more clear and better structurated.